# Qualitative insights on sexual health counselling from refugee youth in Bidi Bidi Refugee Settlement, Uganda: Advancing contextual considerations for brief sexuality-related communication in a humanitarian setting

Miranda G. Loutet[1]*, Carmen H. Logie[2,3,4,5], Moses Okumu[6,7], Madelaine Coelho[8], Karel Blondeel[9,10], Alyssa McAlpine[2], Frannie Mackenzie[3], Simon Odong Lukone[11], Nelson Kisubi[11], Jimmy Okello Lukone[11], Atama Malon Isaac[12], Peter Kyambadde[13,14], Igor Toskin[10]

1 Dalla Lana School of Public Health, University of Toronto, Toronto, Canada, 2 Factor-Inwentash Faculty of Social Work, University of Toronto, Toronto, Canada, 3 Women's College Research Institute, Women's College Hospital, Toronto, Canada, 4 United Nations University Institute for Water, Environment, and Health, Hamilton, Canada, 5 Centre for Gender & Sexual Health Equity, Vancouver, Canada, 6 School of Social Work, University of Illinois at Urbana Champaign, Urbana, Illinois, United States of America, 7 School of Social Sciences, Uganda Christian University, Mukono, Uganda, 8 Department of Sociology, University of Toronto, Toronto, Canada, 9 Faculty of Medicine and Health Sciences, Ghent University, Ghent, Belgium, 10 Department of Sexual and Reproductive Health and Research, UNDP-UNFPA-UNICEF-WHO-World Bank Special Programme of Research, Development and Research Training in Human Reproduction (HRP), World Health Organization, Geneva, Switzerland, 11 Uganda Refugee and Disaster Management Council (URDMC), Arua, Uganda, 12 Yumbe Regional Referral Hospital, Yumbe, Uganda, 13 National AIDS Coordinating Program, Ugandan Ministry of Health, Kampala, Uganda, 14 Most at Risk Population Initiative (MARPI), Kampala, Uganda

* miranda.loutet@mail.utoronto.ca

**Editor:** Sebastian Suarez Fuller, University of Oxford Nuffield Department of Clinical Medicine: University of Oxford Nuffield Department of Medicine, UNITED KINGDOM OF GREAT BRITAIN AND NORTHERN IRELAND

## Abstract

Characteristics of enabling healthcare environments to support brief sexuality-related communication (BSC) are understudied in humanitarian settings. We implemented a qualitative study with refugee youth aged 16–24 living in Bidi Bidi Refugee Settlement to understand the feasibility of implementing BSC in a humanitarian context. We examined feelings toward doctor's visits in general, including types of conversations youth engage in with healthcare providers, as well as comfort, safety, and willingness to talk with healthcare providers about sexual health. We implemented four focus groups with refugee youth in Bidi Bidi, two with young women and two with young men, and applied thematic analysis informed by a social contextual theoretical framework that explores enabling environments for sexual health promotion. Participants (n = 40; mean age: 20 years, standard deviation: 2.2; women: n = 20; men: n = 20) reported relational, symbolic, and material dimensions of context considered important when discussing sexual health. *Relational contexts* included a) trusting relationship with local healthcare practitioners, including practices that foster comfort and confidentiality, and b) family, friends, and mentors as additional sources of health information.

**Data Availability Statement:** The University of Toronto Research Ethics Board (REB) states that qualitative data should not be shared beyond the research study team due to the possibility of participant identification in this highly sensitive setting of a humanitarian context. This has been approved by not only the University of Toronto REB but also the two Ugandan REBs. Please contact University of Toronto's REB for further information about data access: dean.sharpe@utoronto.ca.

**Funding:** This work was supported by ViiV Healthcare Limited (Grant#628520-1652450711). CHL is also funded by the Canada Research Chairs programme (#Tier 2), Canada Foundation for Innovation (#JELF) and the Ontario Ministry of Research and Innovation (ERA). This work was also funded by the UNDP-UNFPA-UNICEF-WHO-World Bank Special Programme of Research, Development and Research Training in Human Reproduction (HRP), a cosponsored programme executed by the World Health Organization (WHO). Funding agencies played no role in the design or execution of the study.

**Competing interests:** The authors have declared that no competing interests exist. Some authors are current or former staff members of the World Health Organization. The authors alone are responsible for the views expressed in this publication and they do not necessarily represent the views, decisions or policies of the World Health Organization.

*Symbolic contexts* refer to values, norms, and beliefs that reflect what is perceived as valuable and worthy, and in turn, what is devalued and stigmatized. Specific to sexual health, participants discussed stigma toward STIs and HIV, devaluation of women in healthcare settings, and generalized fear of doctors and disease as barriers to engaging in dialogue about sexual health with healthcare providers. *Material contexts* include agency linked with resource access and experiences. Youth narratives revealed that positive experiences accessing medication to manage pain and infections increased their willingness to engage in healthcare discussions, whereby clinic layouts and dynamics that compromised confidentiality and privacy reduced the likelihood of sexual health dialogue. Language barriers and healthcare provider time constraints were additional factors that reduced healthcare engagement. Taken together, findings can inform BSC implementation strategies that consider the inner and outer settings that shape sexual health dialogue and sexual health and wellbeing among refugee youth living in humanitarian settings.

## Introduction

By mid-2022 the number of forcibly displaced people worldwide grew to 103 million [1]. Low and middle-income countries hosted 74% of the world's refugees and displaced peoples [1]. Uganda is the fourth largest refugee hosting nation in the world [1] and contains one of the largest refugee settlements, Bidi Bidi, that is home to over 210,000 residents, of which one-quarter are youth and adolescents (age 15–24 years) [2]. Sexual and reproductive health information and service needs are more likely to be unmet among populations experiencing greater vulnerability, including among others adolescents, those with low socioeconomic status, forcibly displaced people, and those living in humanitarian crisis contexts [3, 4]. Interventions are needed to increase acceptability, accessibility and utilisation of sexual and reproductive health services by these populations [5].

Sexuality-related communication has been identified as an area needing urgent attention as health workers may lack the necessary training and knowledge to feel comfortable addressing sexual health concerns and providing friendly, equality-based and confidential services [6–12]. In 2015, the World Health Organization (WHO) published recommendations for brief sexuality-related communication (BSC) based on evidence from case studies, systematic reviews, and a technical consultation with experts in developing programs or offering clinical services to promote sexual health and well-being [12, 13]. BSC aims to promote reciprocal and bidirectional conversation between a health user and provider to come to a health solution opposed to top-down hierarchical communication and health decision. BSC also aims to provide a wholistic-health solution that promotes the health user's autonomy for their health as opposed to 'disease-centered' care [12, 14, 15]. BSC is conceived to be integrated into primary health care services, by training primary health workers (physicians, nurses and others) in sexual health knowledge and BSC skills [16, 17]. By initiating discussions through BSC, health workers can promote sexual health and rights and motivate health users in pursuing a sexual healthy and pleasurable lifestyle, which may also prevent sexually transmitted infections (STIs), including HIV, unintended pregnancies, and address harmful sexual practices such as sexual violence.

BSC uses opportunistic counselling skills and can be used in encounters with less certainty about the duration or continuity of the encounter, such as at the primary care level in

humanitarian crisis settings where encounters are often the first and only point-of-care [12, 18–20]. The "information, motivation and behaviour" model and open-ended approach to questioning of BSC allows health care providers to consider various aspects of sexual health including psychological, biological, well-being, and social dimensions [21, 22]. This may seem difficult to implement in certain primary-care settings with time-constraints, but well trained providers may be quick to internalize the steps proposed for such communication, without prolonging the clinical encounter very much [17, 23]. Another key dimension of BSC is the rights-based approach to include informed consent [19], autonomy [24], and a commitment to confidentiality to remove stigma, discrimination, and stereotypes from sexual health services and in turn, increase the likelihood of achieving desired sexual health outcomes [25]. Low- and middle-income countries and humanitarian settings experience both challenges and opportunities for the implementation of BSC due to the limited health services but greater vulnerability for poorer sexual health [5, 26, 27]. Substantially more work needs to be done to study and validate BSC techniques in different contexts, such as in settings with limited resources (e.g., humanitarian contexts) and among populations in particularly vulnerable situations such as adolescents [27].

Socio-contextual theory [28] can be leveraged to assess how contextual factors shape the feasibility and acceptability of implementing BSC among refugee youth and adolescents in humanitarian settings. Three dimensions of context may shape healthcare relationships required for BSC to be implemented: *relational contexts*, which include relationships with healthcare providers, families, peers, and communities; *symbolic contexts*, which emphasize the social norms and cultural values which shape health-related perspectives; and *material contexts*, referring to resource access and economic opportunities that are the result of systemic factors which may facilitate or restrict healthcare access [28]. Previous literature has shown the importance of these socio-contexts for engaging youth with HIV testing and prevention services in urban humanitarian settings. Among urban refugee youth in Kampala, Uganda, relational contexts through informal (intimate partners and family members) and formal (peer educators and professionals) sources have shown to shape HIV testing experiences, and professional support was key for accessing and utilizing HIV testing and mental health services [29]. Narratives from refugee and displaced youth from the Democratic Republic of Congo, Rwanda, Burundi, and Sudan identified barriers to HIV testing and prevention engagement that included material (e.g., transportation costs and language barriers) and symbolic (medical mistrust and inequitable gender norms) contextual factors [30].

To inform the implementation of future sexual health interventions such as BSC in humanitarian settings, we must better understand the current conditions and contexts that shape refugee youth's engagement with the healthcare system, such as logistic barriers restricting healthcare access, availability of sexual health resources, and youth's perceptions of the healthcare practitioners providing services. Implementing BSC among refugee youth in Uganda would require sensitivity to the socio-contextual factors that shape sexual health practices. Therefore, we aimed to understand how refugee youth aged 16–24 living in Bidi Bidi Refugee Settlement felt towards doctor's visits in general, as well as comfort, safety, and willingness to talk with healthcare providers about sexual health.

## Methods

### Overview

We conducted a community-based research study in Bidi Bidi Refugee Settlement in collaboration between the University of Toronto, the Uganda Refugee and Disaster Management Council (URDMC), and the Ugandan Ministry of Health. The study reported here used a

qualitative study design with focus groups of refugee youth conducted from August 21–27 2021 and findings from a thematic analysis are presented (Consolidated criteria for reporting qualitative research checklist in Supporting Information (S1 Checklist)). It was a sub-study of the primary study called Todurujo na Kadurok (Empowering Youth), which was an exploratory sequential mixed-methods design that started with a qualitative phase to inform a randomized controlled trial with HIV self-testing kits for refugee youth [31]. The primary aim of the qualitative phase of the Todurujo na Kadurok (Empowering Youth) study was to understand HIV testing needs and experiences. Additionally, within the qualitative phase of Todurujo na Kadurok, we included a series of questions used in previous research [17, 23, 32] to better understand the feasibility and acceptability of sexual health counselling in clinics to inform implementation and enabling environments for BSC, results from which are presented in this paper.

## Setting and participants

The Bidi Bidi Refugee Settlement, Yumbe district in Northwestern Uganda was established in September 2016 and is the second largest refugee settlement in Uganda with over 210,000 residents; one-quarter are youth and adolescents aged 15–24 and the majority (>99%) are from South Sudan [2, 33]. In Bidi Bidi sexual health services are delivered through health centres that are distributed throughout the Zones of the settlement, staffed by doctors and nurses, and funded by the government or non-governmental organizations. Refugees can either be referred to sexual health services through primary health care or access it directly. In addition, there are sexual health officers employed to provide sexual and reproductive health throughout each Zone. Study design and implementation was conducted in collaboration with the Bidi Bidi organization, Uganda Refugee and Disaster Management Council (URDMC). URDMC and Ugandan Ministry of Health provided approval of study proposal and protocol.

Inclusion criteria for the focus groups participants were refugees aged 16–24 years who live in Bidi Bidi, and who can speak English, Bari and/or Juba Arabic. Focus groups were separated by gender (two groups of young women and two groups of young men) to increase participants' feeling of comfort and confidentiality to discuss sensitive topics, with 10 participants per group. Participants were recruited through word of mouth using purposive sampling by URDMC staff and peer navigators (PN) who were themselves refugees aged 20–30 living in Bidi Bidi and fluent in English, Bari and/or Juba Arabic.

## Data collection

Four focus groups were facilitated by two URDMC research assistants (one woman and one man leading the respective gender groups), supervised by URDMC research staff, and supported by two PNs providing real-time translation. Focus group discussions were conducted in a private room within the community, which was easily accessible to all participants, and took on average two hours. All staff received training on qualitative research methods including focus group facilitation skills, confidentiality, and psychological first aid. Focus groups were audio recorded, transcribed verbatim (any Bari or Juba Arabic translated to English in transcripts) by URDMC research staff.

We used a semi-structured focus group guide to facilitate the focus group discussions. As part of the development of the focus group guide, the URDMC research assistants piloted the guide and provided feedback to ensure questions were clear and relevant to the study population. First, participants were asked "*How do you generally feel about doctor's visits*?" Participants were further probed to discuss the kinds of conversations they have when they go to the doctor, whether they can discuss sexual health with their doctor, barriers and facilitators that

exist to having conversations about sexual health, and who else they can talk to about sexual health. Participants were then asked about their feelings towards talking to providers about sensitive topics related to sexual health: 1) "*How **comfortable** do you think that people in your community (people like you) would be talking to providers about sensitive topics related to sexual health*?"; 2) "*How **safe** do you think that people in your community (people like you) would be talking to providers about sensitive topics related to sexual health*?"; and 3) "*How **willing** do you think that people in your community (people like you) would be talking to providers about sensitive topics related to sexual health*?" For each of these questions, participants were probed further to ask why people would or would not feel comfortable/safe/willing to talk to providers about sensitive topics related to sexual health and what would help the participant or others in the community to feel more comfortable/safe/willing to talk to providers about sensitive topics related to sexual health.

## Data analysis

A codebook was created based on the socio-contextual theory that conceptualizes health-enabling environments as interconnected material, symbolic, and relational contexts [28]. We used the Dedoose software [34] to code transcripts. We approached thematic analysis in both an inductive way such that code and theme development was directed by the content of the data and in a deductive way by relating codes to relevant themes within each context of the socio-contextual theory [28, 35]. We applied this theoretically flexible approach of thematic analysis [35, 36] to explore specific aspects of material, symbolic, and relational contexts that support or prevent refugee youth from engaging with sexual health services and having conversations about sexual health. We chose to use this approach due to the flexible nature, allowing us to consider how barriers and facilitators to sexual health services and conversations are impacted by different contexts. Two University of Toronto research assistants (MGL and MC) coded the data independently and discrepant codes were reviewed together with a third party (CHL) when necessary. All codes and themes were reviewed by the senior researchers (CHL, MO), with local input from URDMC research staff (SOL, NK), and expertise from agency collaborators (KB, IT).

## Ethics

We obtained ethical approval from Mildmay Uganda Research Ethics Committee (REC REF 0802–2021), Uganda National Council for Science and Technology (SS884ES), and the University of Toronto Research Ethics Board (37496). URDMC and Ugandan Ministry of Health provided letters of support for ethics submissions. All participants were given details about the study purpose and activities, and provided written informed consent prior to participation. All participants were asked to provide consent to participate in the study for themselves, regardless of age, in line with the: 1) community collaborator recommendations, 2) Uganda's HIV and AIDS Prevention and Control Act (2014) that acknowledges the agency of youth as young as 12 to make the decision independently to have an HIV test [37], 3) studies that acknowledge that youth as young as 14 years can provide informed consent to participate in research [38–41], and 4) approved by Ugandan and University of Toronto ethics boards.

## Inclusivity in global research

Additional information regarding the ethical, cultural, and scientific considerations specific to inclusivity in global research is included in the Supporting Information (S2 Checklist).

## Results

Young women (n = 20) and men (n = 20) were engaged across four focus groups. Participants ranged from 16 to 24 years old (mean age = 20, standard deviation (SD): 2.2), were mainly students (87%), with a total of 40% having less than secondary school level of education, 53% completed secondary school, and 7% attended some college or above. We will describe the following themes within each socio-context that emerged from the focus groups: *relational contexts* (relationships with local healthcare practitioners, family and friends, and other community providers), *symbolic contexts* (feelings of stigma relating to STIs and HIV, devaluation of women in healthcare encounters, and generalized fear of doctors and disease), and *material contexts* (access to treatment, clinic's layout and dynamics, and healthcare providers' communication and availability) that support or prevent refugee youth from engaging with sexual health services in Bidi Bidi and having conversations about sexual health.

### Relational contexts—How relationships contribute to access and usage of sexual health services and communication

**Relationships with local healthcare practitioners.** Relational contexts included trusting relationships with local healthcare practitioners particularly in primary health care that then refer them to sexual and reproductive health services, including practices that foster comfort and confidentiality.

Participants discussed how relationships with healthcare professionals can facilitate meaningful communication with patients, such as through encouraging patients to access support. In addition, participants can receive individualized medical knowledge from healthcare professionals to make sense of individual sexual health scenarios. As one young refugee noted, "I can go and discuss sexual health with the doctor because as a youth, I may have unprotected sex with my girlfriend, then I can discuss it with the doctor, to avoid other problems in the future like maybe the girl getting pregnant" (man, age 19, focus group number (FG#) 2, identification number (ID#) 7). Youth trust the information that they receive from local healthcare practitioners, which makes them more likely to view their experience at the clinic as positive and feel empowered:

> I went to this health center here, (blinded), to test for HIV. [The doctor] gave me condoms and they asked me if I had used it before. I said 'no', they got an artificial penis and they illustrated for me how I can put the condom, so I was happy because they told me that when I am putting [it on], I should leave space for collecting sperm so I got knowledge and I was happy

(man, age 21, FG#4, ID#3).

Some participants discussed how health care providers can support youth with making decisions about their sexual health. For example, one young man refugee identified this hierarchical but trusting relationship, "I can discuss with the doctor about family planning and sexual health because they're doctors . . . They will guide you on how to use condoms, then you will get [to] know the procedures of using a condom" (man, age 18, FG#2, ID#1).

Healthcare providers were also sources of emotional support, predicated on assumed confidentiality between refugee youth and doctors. A young woman refugee explained how her relationship with the doctor was comforting: "You feel comfortable when you know that the person you are talking to is a doctor who cannot spread issues anyhow. There you can talk without fear not developing, that if I say something this person will not take it anywhere" (woman, age 20, FG#1, ID#9). Refugee youth reported that healthcare providers were attentive

to their needs and created a comfortable environment. For example, a young refugee explained, that "if I am not comfortable with the place, then I will tell the doctor that, 'please, I am not comfortable in what I am going to express', then after that, the doctor will make the place comfortable" (woman, age 21, FG#3, ID#2).

**Support from friends, family, and partners.**   Participants identified friends, family, and partners as additional sources of health information and as sources of social support, such as through accompaniment to healthcare visits. Specifically, mothers were described as an initial point-of-contact for healthcare matters for young girls. A young woman refugee describes: "for us, the girls, if you may be having some pain in your lower abdomen, you can approach your mother" (woman, age 20, FG#1, ID#8). Another participant explains that "I can talk to my mum because mum is your trusted friend, and they can give you advice on how to live a safe life" (woman, age 21, FG#3, ID#2). Participants also identified their partners as sources of information and feeling secure about their health, as such, a young woman refugee said "I can talk to my boyfriend such that we can get to know our health and in such a way we can keep safe" (woman, age 18, FG#3, ID#1).

Some participants discussed how receiving support from friends can potentially facilitate healthcare access. One young woman described her experiences as a support person for her friend:

*I had a friend who developed candida, and she feared to go to the doctor because they were all male doctors, so I went with her, and it was me who was explaining to the doctor so I feel with friends, it can be ok, because they can help you talk to the doctor instead of die quietly alone*

(woman, age 23, FG#3, ID#9).

Similarly, another young woman refugee illustrated the positive impact of friend support when visiting a health centre "for me friend is better because as you move to the health centre with your friend [they] will be just advising and counselling you and when I reach there, I see that I cannot explain, I will ask [my friend] to help me out, and the doctor will ask who is suffering from that sickness, then I will say it's me, and I [am] with friend because I have fear" (woman, age 19, FG#3, ID#5).

Among young refugees who described experiencing mistrust of healthcare providers, they discussed turning to family and friends to entrust their healthcare concerns, which demonstrates the importance of relationships that emphasize confidentiality about health: "I can talk to one of my brothers in the family because not all brothers are okay, some are not okay, some are okay, so you get the one you trust to keep secret, then you share the talk with him" (man, age 20, FG#4, ID#2). Another young man also discussed trusting their friend with upholding confidentiality of their sexual health information: "I think in the community the person I can talk to about sexual health is my closest friend with whom I know that if we speak our things they are so confidential" (man, age 23, FG#4, ID#9).

**Additional access through community resources.**   Refugee youth also identified additional community resources where they can access health care and talk about sexual health. One refugee youth said "if I don't go to the doctor there are also health workers who work in the community and I can go to them and explain my condition they can help me (man, age 20, FG#2, ID#5)" and another said "for cases of rape and defilement I can go the chairman of the village and share with him (man, age 23, FG#2, ID#3)." Multiple refugee youth brought up accessing sexual health care through Village Health Teams (VHTs) which are volunteers who provide basic health care services to local communities: "for me I would communicate through the VHT that I trust" (woman, age 17, FG#3, ID#6)

## Symbolic contexts—Social norms and cultural values that impact access and usage of sexual health services and communication

**Feelings of stigma relating to STIs and HIV.** The stigma of sexual health issues was discussed by many participants, which shaped their level of comfort related to accessing healthcare and treatment. Participants described that they often feel stigmatized by healthcare staff and practitioners when they talk about their sexual behaviour. One young man refugee described such an experience with his doctor: "when you reach the doctor for HIV test, they will ask you how many girlfriends do you have and it is always very difficult to respond because you may be having more than five . . . If I say five, I will appear to be a womanizer, so I did not like it" (man, age 22, FG#4, ID#7). Other participants referred to feeling irresponsible or not trusted with sexual health information in the community: "here, when you talk about sexual health, they don't advise people about that because they think that if they advise, especially tell ladies about this, they will go and spread the knowledge not to have a lot of care" (woman, age 20, FG#1, ID#9). Even when doctors appeared to be non-judgmental, participants described that merely visiting a clinic may draw negative attention and result in the spread of rumors. For example, one young refugee explained that "if you go there to be counselled about HIV, if you are talking there beyond one hour, people may be thinking that this man is positive, yet you are negative . . . so the more time you take makes people to start thinking different things" (man, age 21, FG#4, ID#3).

**Devaluation of women in healthcare encounters.** Participants described that women within the community experience unique barriers such as experiences of harassment and over-sexualization, which can create an environment that makes women and girls reluctant to pursue healthcare. As one young woman refugee notes, "when I approach a doctor, I don't want the doctor to say you are beautiful . . . [it] is not concerned about why I have come to him" (woman, age 20, FG#1, ID#9). Another young woman expressed fear during interactions with healthcare providers due to the experience of inappropriate comments: "when the doctor starts the nonsense like you are beautiful, I want your phone number, you develop fear" (woman, age 23, FG#1, ID#3). These negatives experiences with healthcare professionals may make women less receptive to accessing healthcare. For example, one young woman refugee expressed hesitancy related to discussing woman-specific issues with doctors who are men: "I really feel bad if a male doctor starts discussing with me about menstruation" (woman, age 22, FG#3, ID#4). It follows that women and girls would be hesitant to access healthcare or to make disclosures of a sensitive nature such as sexual violence or abuse without access to and relationships with doctors who are women in a setting that they feel comfortable. A young woman refugee explained: "if I am not comfortable with the place then I will tell the doctor that please I am not comfortable in what I am going to express, then after that the doctor will make the place comfortable for you then you can start explaining to [them] your issue for example I was raped when I went to fetch firewood" (woman, age 21, FG#3, ID#2).

**Generalized fear of doctors and disease.** Beliefs related to healthcare, regardless of their accuracy, may disincentivize an individual from seeking out health-related support. Participants described fear of healthcare providers due to being overwhelmed with possible health issues during a single doctor's visit: "for me sometimes, I fear, because when you reach the doctor like this, they will want to test you for both malaria and HIV even if you are not ready yet" (man, age 18, FG#2, ID#1) and "when you visit the doctor at that particular moment, you feel so scared, more so if it is your first time to visit the doctor. And some doctors can make you feel so scared because they will first isolate you and tell you that your issue is different, and this can be very scary" (man, age 20, FG#2, ID#8).

Furthermore, some refugee youth held the belief that doctors only deliver 'bad news', which may make individuals hesitant to seek out healthcare services. For example, one participant described how seeking healthcare services related to HIV and STIs can be especially concerning given fatalistic perceptions related to HIV: "It is very scary because once you have entered there, by the time the doctor is counselling you, he will ask you that 'now that you have come for the test, if we find that you are positive, how are you going to feel? What are you going to do? In your mind, you feel that you already have the disease with you. You will think that maybe the doctor has seen something in me so that really scares" (man, age 23, FG#4, ID#9).

## Material contexts—Systemic factors that impact access and usage of sexual health services and communication

**Treatment access.** Youth narratives revealed that positive experiences accessing medication to manage pain and infections increased their willingness to engage in healthcare discussions. Participants shared stories of meaningful and helpful experiences with doctors that provided reassurance and immediate treatment: "there is a time when I visited the doctor when my penis was swollen, and since I entered [the clinic], I felt very happy because I was badly off, but the doctor convinced me that . . . I am going to get better. He gave me some drugs and the swelling was reduced" (man, age 21, FG#4, ID#4). The availability of medication and tangible resources encourages refugee youth to visit clinics and receive formal healthcare. One young woman refugee describes this experience in relation to access to menstruation services: "we ladies our periods at times pains more than even the normal pain . . . and you [go to the health centre] and explain to them, then they will be in the position to look for a drug that will help you" (woman, age 16, FG#1, ID#6).

**Gender-specific services.** Young women refugees highlighted that when there are women practitioners available at the health centres, women and girls are more likely to seek out health-related support: "ladies should approach a female doctor . . . so that they will address the issues freely because for men their issues for ladies they don't know now. If you tell, they say 'eeeh kumbe', this is happening with 'so and so', now at least if it's a fellow woman, [they] will say this is normal" (woman, age 20, FG#1, ID#9). They also would rather speak to a nurse who is a woman if a doctor who is a woman is not available: "Topics about menstruation I really feel bad if a male doctor starts discussing with me about menstruation. At least there should be a female doctor if not a female nurse." (woman, age 22, FG#3, ID4)

**Private spaces.** Experiences with a clinic's layout and dynamics that compromise confidentiality and privacy reduced the comfortability of engaging in sexual health dialogue and utilization of services among participants. For example, a young man refugee said: "sometimes the counselling door may be near the patients so if you are going for your test in our health centre you find that somebody is just seated nearby the door, and you are there, so when you look at [other patients] you will have fear because they are going to expose you, so this can make you hide a lot from the doctor" (man, age 23, FG#4, ID#6).

Participants also described that confidentiality must be guaranteed not only by the doctors, but all healthcare staff, including translators:

*The major concern generally in our setting here, we have translators, so if the translators would be trained very well, so that they can learn to keep secrets, they don't expose whatever you have said, there I think it would help. If there is no confidentiality kept there, you will find it very hard so that is why sometimes people don't tell the health providers what is happening*

(man, age 24, FG#4, ID#8).

Many youth participants provided solutions for increasing confidentiality in the health centres. One youth shared "if the counselling room [is] far from other patients, I think it can motivate people to go and talk to the doctor what is disturbing them, and then also if the doctor is trustworthy, loyal, and maintain confidentiality, can also motivate other people to go and share with him [and] express what they are feeling" (man, age 24, FG#4, ID#8).

**Healthcare provider communication.**   Language barriers and healthcare provider time constraints were additional factors that reduced healthcare engagement. While translators could alleviate the difficulties related to language barriers, this created additional barriers related to confidentiality.

*In our place here, it is very common for the doctors to understand and use English while our people here do not know English so if they are to communicate to the doctor, they need a translator, and the translator is someone that we know in the community. So, if I have sensitive stories to tell the doctor, I may not share it freely because I will be fearing the translator that he/she may expose this in the community*

(man, age 22, FG#4, ID#7).

**Healthcare provider availability.**   The availability of doctors are additional concerns of refugee youth. One young refugee said: "When I visit the doctor, sometimes one of the barriers to having this conversation is that sometimes the doctors are so busy they don't give you the time" (man, age 20, FG#2, ID#5). When asked what length of time for this type counselling would be ideal, refugee youth answered between 30 minutes and 2 hours, demonstrating how needs and comfort levels may differ by person: "for 30 minutes will be enough, I feel it will be enough for me" (woman, age 17, FG#3, ID#6) and "I will need like 2 hours so that it will be enough for me to learn very many things and also for him to counsel me enough" (woman, age 18, FG#3, ID#8).

## Discussion

Narratives from refugee youth in Bidi Bidi refugee settlement identified socio-environmental factors that produced either enabling environments or barriers to engaging with sexual health services and having sexual health conversations in the settlement. Relational contexts provided an enabling environment for sexual health services through trusting relationships with local healthcare providers, seeking support from family, friends, and partners, and seeking additional services in the community. Symbolic barriers to engaging in sexual health services included social norms and cultural values that felt stigmatizing in relation to STIs and HIV by others in the community or healthcare providers, devaluation of women in healthcare encounters, and generalized fear of doctors and disease. Material contexts created enabling environments through increasing access to treatment, but also provided barriers to engagement through the clinic's layout and dynamics compromising confidentiality and privacy, language barriers, and healthcare provider time constraints. Themes emerging from each of these contexts demonstrated the overlapping complexity of relationships, stigma, cultural norms, and material factors impacting refugee youth's experiences with sexual health services. BSC, and other sexual health communication interventions more broadly, have been shown to increase STI prevention behaviours and reduce STI incidence [13, 21, 42–49], but little research has explored the facilitators and barriers to creating a health-enabling environment for such interventions in diverse settings and populations.

Other studies, including a systematic review of sexual health communication between healthcare providers and adolescents and young adults, consistently identify confidentiality as an important factor to foster clinician-adolescent sexual health communication [50–53] and is an important component of BSC to create reciprocal and bidirectional conversations between a health user and provider, along with using a rights-based approach to services. Confidentiality was a central theme that played out in *relational* (i.e., relationships with healthcare providers) and *material* (i.e., clinic's layout and dynamics) contexts among refugee youth in our study. Adolescents from a primary care youth-centered community health clinic in the United States (US) also described experiences of perceived judgement from clinicians during HIV/STI communication [50], which refugee youth in our study also identified as a *symbolic* context barrier to engaging with sexual health services due to STI and HIV stigma perpetuated by healthcare providers. Most adolescents included in that US study were women, and similarly to the young women in our study, there was an experience of judgement and stigma, which was illustrated by participants as being devalued during healthcare encounters. When considering implementation of BSC in our setting, the primary aim of BSC to have a commitment to confidentiality to remove stigma, discrimination, and stereotypes from sexual health services is crucial. Narratives from our study show that refugee youth who are uncomfortable with healthcare providers will seek health information and healthcare accompaniment from family and friends, which aligns with reports from high school students in the US that ranked partners and friends as the major source of sexual information over healthcare professionals [54]. Peer engagement is a key component of youth HIV prevention and care cascades [29, 55–61] and therefore may be successfully integrated within future BSC interventions. BSC uses opportunistic counselling skills and can be used in encounters with less certainty about the duration or continuity of the encounter, therefore the fact that refugee youth in Bidi Bidi described accessing services with the support of friends could pose an opportunity for healthcare providers to use BSC among more than one youth at a time. Findings from our study can provide a framework with interconnected relational, symbolic, and material contexts to inform BSC implementation strategies.

This is among the first studies to assess the feasibility and acceptability of implementing BSC for adolescents in a humanitarian setting. The WHO explicitly stated the importance of evaluating BSC in different contexts such as humanitarian settings that have particular facilitators and barriers to sexual health interventions that need to be understood to facilitate impact and sustainability [12]. A systematic review of sexual health utilization in humanitarian settings found interpersonal and peer-led education interventions were the most effective to increase service utilization of HIV, STI, and maternal health services [5], which was similar to the *relational contexts* with healthcare providers, family, friends, and community resources described as facilitators of sexual health services in our study. Another systematic review of the effectiveness of sexual health interventions in humanitarian settings found evidence to support interventions that included home visits and integration of HIV and sexual and reproductive health services [26]. This systematic review did not include any studies focusing on adolescents, which may reflect why these interventions are counter to what was acceptable to youth in Bidi Bidi; there may be issues with confidentiality during home visits, impacts of community stigma, and/or fear of doctors and disease if too many health issues are integrated into a single doctor's visit. These barriers to accessing sexual health services in Bidi Bidi highlight the need for specialized training for healthcare providers to provide wholistic-health solutions to promote the health user's autonomy if BSC was to be introduced in this setting. Furthermore, young people living in humanitarian settings are at a higher risk of mental health disorders, which further compounds risks for poor sexual and reproductive health outcomes, and impacts sexual and reproductive health service use [27, 62]. An important component of BSC

is providing care beyond the prevention of STI and unintended pregnancy to include sexual well-being and refugee youth in Bidi Bidi spoke about how positive relationships with health-care providers that encouraged them to seek sexual health services also provided emotional support and skills to engage in sexual health promotion strategies such as condom use, while also increasing empowerment. Therefore, future studies of the feasibility and acceptability of BSC in Bidi Bidi should integrate both mental and sexual health principles and services. This demonstrates the need for rigorous implementation research that is specific to the crisis setting and population.

This study also provides insights for implementation logistics for BSC in humanitarian settings where resources are scarce. BSC is meant to be integrated into primary health care services through training of health workers. Therefore, another opportunity for BSC to be successfully implemented in humanitarian settings would be to also engage and train the Village Health Teams, which refugee youth in Bidi Bidi reported to use often for sexual health services, and nurses who are women to provide gender-specific care. This would also help in primary-care settings with time constraints because refugee youth in Bidi Bidi reported that doctors did not have enough time to speak with them. This appears to be feasible in settings already stretched for resources as demonstrated by feasibility studies of BSC in Peru and Moldova, where healthcare providers were both willing and committed to learning new techniques and strategies to implement BSC, improve their knowledge and communication skills, and the quality of care [17, 23]. Furthermore, the diverse needs of refugee youth–exemplified through the varying amount of time Bidi Bidi youth reported that they wanted for sexual health counselling–can be met by BSC's "information, motivation and behaviour" model and open-ended approach to questioning. However, important lessons from BSC feasibility studies with health-care providers stress the importance of structural factors including integration of priority setting across multiple levels (provider, client and institution), rather than solely focussing on training providers, in order to successfully implement BSC [17, 23]–which will be a challenge in humanitarian settings.

## Limitations

There were several limitations to our study. The use of focus groups may have provided a barrier to individual participants sharing personal experiences, and the purposive sampling of focus group participants may have resulted in a study population that is already more engaged with community services than the general residents of Bidi Bidi. Due to the scope of the primary study, only refugee youth were included in focus groups; however, to fully understand the extent of barriers and facilitators to sexual health counselling would have required perspectives from various healthcare providers, including understanding the extent of sexual health-related stigma that exists in Bidi Bidi. Inclusivity in relation to sexual and gender diversity was not explored in the current study, although it is an important aspect of sexual health communication among young people [51]. Furthermore, as this qualitative study was embedded within a larger mixed-methods study on HIV testing, BSC was not the primary aim of the study and more details relating to the implementation and impact of BSC was not feasible in the present study. However, this study provides preliminary data to inform future implementation studies of BSC for youth in humanitarian settings such as Bidi Bidi Refugee Settlement.

## Conclusions

Although there is evidence that BSC increases sexual health promotion behaviors and reduces STI incidence [13], substantially more work needs to be done to understand practical considerations for implementation of BSC in diverse settings, populations, and providers [12]. Our

study provides a framework of interconnected socio-environmental factors to inform implementation strategies for BSC among refugee youth in the Bidi Bidi Refugee Settlement. It is clear from comparisons to other studies of BSC, and sexual health communication more broadly, that context specific implementation studies are crucial to understanding the feasibility and acceptability of such interventions in various crisis settings. Together these findings reveal the need for a BSC intervention that integrates clinical and community-based factors including peer engagement and healthcare provider training to instill principles of confidentiality, gender equality, and human rights to increase trust and dismantle gender discrimination and other forms of stigma and discrimination. Implementation studies of BSC in humanitarian settings provide an opportunity to incorporate psychological services alongside sexual health communication to explore the role of BSC in addressing human rights and overall sexual well-being. Findings can inform BSC implementation strategies that consider the complex contexts that shape sexual health dialogue and sexual health and well-being among refugee youth in humanitarian settings.

## Supporting information

**S1 Checklist. COREQ (COnsolidated criteria for REporting Qualitative research) checklist.**
(PDF)

**S2 Checklist. Inclusivity in global research.**
(DOCX)

## Acknowledgments

We would like to thank the research assistants who facilitated the focus groups, peer navigators for supporting the study, and participants for their valuable contributions to this study.

## Author Contributions

**Conceptualization:** Miranda G. Loutet, Carmen H. Logie, Moses Okumu, Karel Blondeel, Simon Odong Lukone, Peter Kyambadde, Igor Toskin.

**Data curation:** Miranda G. Loutet, Madelaine Coelho, Alyssa McAlpine, Frannie Mackenzie, Simon Odong Lukone.

**Formal analysis:** Miranda G. Loutet, Madelaine Coelho.

**Funding acquisition:** Carmen H. Logie, Moses Okumu, Peter Kyambadde.

**Investigation:** Miranda G. Loutet, Carmen H. Logie, Moses Okumu, Alyssa McAlpine, Frannie Mackenzie, Simon Odong Lukone, Nelson Kisubi, Jimmy Okello Lukone, Atama Malon Isaac, Igor Toskin.

**Methodology:** Miranda G. Loutet, Carmen H. Logie, Moses Okumu, Alyssa McAlpine, Simon Odong Lukone, Nelson Kisubi, Igor Toskin.

**Project administration:** Miranda G. Loutet, Alyssa McAlpine, Frannie Mackenzie, Simon Odong Lukone, Nelson Kisubi, Jimmy Okello Lukone, Atama Malon Isaac.

**Supervision:** Carmen H. Logie, Moses Okumu, Karel Blondeel, Igor Toskin.

**Writing – original draft:** Miranda G. Loutet, Madelaine Coelho.

**Writing – review & editing:** Miranda G. Loutet, Carmen H. Logie, Moses Okumu, Karel Blondeel, Alyssa McAlpine, Frannie Mackenzie, Simon Odong Lukone, Nelson Kisubi, Jimmy Okello Lukone, Atama Malon Isaac, Peter Kyambadde, Igor Toskin.

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
