## [Decision Letter · Decision Letter 0]

25 Jun 2024

PONE-D-23-42992Qualitative insights on sexual health counselling from refugee youth in Bidi Bidi Refugee Settlement, Uganda: advancing contextual considerations for brief sexuality-related communication in a humanitarian settingPLOS ONE

Dear Dr. Loutet,

Thank you for submitting your manuscript to PLOS ONE. After careful consideration, we feel that it has merit but does not fully meet PLOS ONE’s publication criteria as it currently stands. Therefore, we invite you to submit a revised version of the manuscript that addresses the points raised during the review process.

Apologies for the delay in review of your article. Please ensure that your revision includes reporting your results inclusive of all aspects of the COREQ reporting standards. Please also attend to the reviewers comments, particularly on alignment of the study aim and reported outcomes, and adjustments to the text to improve reader understanding of the application of theory to your findings. All rebuttals (i.e. disagreements) to reviewer comments must be fully justified, but will be taken into account in review of your resubmission. 

We look forward to receiving your revised manuscript.

Kind regards,

Sebastian Suarez Fuller, PhD

Academic Editor

PLOS ONE

3. In the online submission form you indicate that your data is not available for proprietary reasons and have provided a contact point for accessing this data. Please note that your current contact point is a co-author on this manuscript. According to our Data Policy, the contact point must not be an author on the manuscript and must be an institutional contact, ideally not an individual. Please revise your data statement to a non-author institutional point of contact, such as a data access or ethics committee, and send this to us via return email. Please also include contact information for the third party organization, and please include the full citation of where the data can be found.

Reviewers' comments:

Reviewer's Responses to Questions

**Comments to the Author**

1. Is the manuscript technically sound, and do the data support the conclusions?

Reviewer #1: Yes

Reviewer #2: Partly

2. Has the statistical analysis been performed appropriately and rigorously? 

Reviewer #1: N/A

Reviewer #2: N/A

3. Have the authors made all data underlying the findings in their manuscript fully available?

Reviewer #1: No

Reviewer #2: Yes

4. Is the manuscript presented in an intelligible fashion and written in standard English?

Reviewer #1: Yes

Reviewer #2: Yes

5. Review Comments to the Author

Reviewer #1: The authors have provided a well written and engaging article providing useful qualitative insights into contextual factors and issues around sexual health counselling and youth health seeking behaviour in a humanitarian setting. This is an important contribution to the literature, and provides useful evidence around the complexities of sexual health needs amongst youth in challenging settings.

The secondary analysis of data has also generated important findings which provide information around the feasibility and implementation of sexual health interventions in such settings. However, I have concerns about the way the data have been analysed and presented. Authors have presented that the primary aim of the data collected was to understand HIV testing needs and experiences. Yet, much of the findings have been presented broadly around youth engagement with the healthcare systems and interactions with specific health care providers within the study context. Further, the authors aim to explore considerations for implementing BSC, the findings presented do not adequately present contextual considerations which are transferable to aspects or components of BSC specifically. For example, could findings around relationships built between providers and youth contribute towards "reciprocal and bidirectional conversations" required in BSC? Does information sourced from family and peers contribute to "recognition of user's autonomy?". How were the findings around time to provide counselling related to feasibility of the opportunistic nature of counselling BSC offers in settings of uncertainty such as refugee camps. The authors have rightly acknowledged the limitations of their data and analysis. However, given that the title, abstract and introduction set the study's focus on BSC, there is need to show how the study findings contribute to BSC rather than sexual health counselling and services generally provided. Overall, the study does not adequately or clearly present findings within considerations for BSC. Perhaps the authors can consider how best to present findings or the discussion to draw out more clearly how the results take into consideration aspects of BSC.

Most of the findings also are related to doctors as the healthcare providers, have the authors considered the role of other providers in this setting (nurses and sexual health officers) towards feasibility and implementation of BSC in refugee and humanitarian settings?

With regards stigma towards HIV, the findings seem to be presented as beliefs of healthcare staff and practitioners rather than perceptions of youth about healthcare practioners. There may be a need to clarify how these findings are presented.

Reviewer #2: Thank you for the opportunity to review your manuscript. This important and understudied subject.

Broad comments:

The introduction is well written and informative.

The methods section requires a complete revision for sentence structure, flow and grammar. Some paragraphs contain sentences that are repetitive or a rewording of something that has already been mentioned.

The content of the methods section requires substantial revisions. Please use some kind of qualitative reporting tool (such as the COREQ) to check your basic qualitative reporting. What is the research design? Why do you use an interview guide for a focus group discussion?

Your data analysis section does not provide enough detail. Tell the reader how and why you applied the theory. How did you check the coding? Who did you discuss and refine codes with? What did this process look like? How did you use thematic analysis? Why is this the appropriate approach to use? https://www.thematicanalysis.net/understanding-ta/

As this is a deductive approach to analysis, it’s really important that you discuss rigor and reflexivity, and that you didn’t simply see what the theory already sees. There are lots of examples of how to do this. Please look at Operationalising rigor (Lincoln and Guba, 1986, Morse, 2015).

Results

There is some overlap between sections in the results which need clarification or clearer descriptions as to why they have ended up in the section they are in. For example, the content of relational contexts is all very positive, with people talking about positive interactions with health care workers. But then in symbolic contexts, you talk about womens negative interactions with health care workers, for example:

“when the doctor starts the nonsense like you are beautiful, I want your phone number, you develop fear” (woman, age 23, FG#1, ID#3). These negatives experiences with healthcare professionals may make women less receptive to accessing healthcare.

I also see this as fitting with ‘relational context’ only that its not a positive interaction. Does your theory only describe relational contexts as positive? If yes, please describe it, otherwise it’s too broad. All of your headings need more description as to what the theory you are using describes. The information in the introduction is not sufficient.

Discussion

Your discussion is also well written and informative, but I think you can do more to tease out the use of your findings particularly for this group, in this setting. How can we use your results to make things better? What are the resource implications? Can you refer to studies or examples of best practice that could be used alongside your findings?

Your limitations are merged with the discussion – its hard to see where the limitations end and the discussion continues. Please amend.

6. PLOS authors have the option to publish the peer review history of their article (what does this mean?). If published, this will include your full peer review and any attached files.

Reviewer #1: No

Reviewer #2: No

---

## [Author Response · Author response to Decision Letter 0]

2 Aug 2024

Response: Thank you for providing those templates, they were very helpful. We have revised the manuscript so it meets PLOS ONE’s style requirements 

Response: Thank you for clarifying this point about informed consent. We had initially included an explanation for why all participants, including minors, were asked to provide their own written informed consent but realize it might have not been clearly stated. We have now clarified in the final sentence of the Methods/Ethics section (page 11, lines 294-301): “All participants were asked to provide consent to participate in the study for themselves, regardless of age, in line with the: 1) community collaborator recommendations, 2) Uganda’s HIV and AIDS Prevention and Control Act (2014) that acknowledges the agency of youth as young as 12 to make the decision independently to have an HIV test,37 3) studies that acknowledge that youth as young as 14 years can provide informed consent to participate in research,38–41 and 4) approved by Ugandan and University of Toronto ethics boards.”

3. In the online submission form you indicate that your data is not available for proprietary reasons and have provided a contact point for accessing this data. Please note that your current contact point is a co-author on this manuscript. According to our Data Policy, the contact point must not be an author on the manuscript and must be an institutional contact, ideally not an individual. Please revise your data statement to a non-author institutional point of contact, such as a data access or ethics committee, and send this to us via return email. Please also include contact information for the third party organization, and please include the full citation of where the data can be found.

Response: The University of Toronto Research Ethics Board (REB) states that qualitative data should not be shared beyond the research study team due to the possibility of participant identification in this highly sensitive setting of a humanitarian context. This has been approved by not only the University of Toronto REB but also the two Ugandan REBs. We will revise the data statement to include the University of Toronto’s REB as the non-author institutional point of contact who can provide further information about data access: dean.sharpe@utoronto.ca

Review Comments to the Author

Reviewer #1: 

Comment 1: The authors have provided a well written and engaging article providing useful qualitative insights into contextual factors and issues around sexual health counselling and youth health seeking behaviour in a humanitarian setting. This is an important contribution to the literature, and provides useful evidence around the complexities of sexual health needs amongst youth in challenging settings.

Response 1: Thank you for your careful review and comments to strengthen our paper. 

Comment 2: The secondary analysis of data has also generated important findings which provide information around the feasibility and implementation of sexual health interventions in such settings. However, I have concerns about the way the data have been analysed and presented. Authors have presented that the primary aim of the data collected was to understand HIV testing needs and experiences. Yet, much of the findings have been presented broadly around youth engagement with the healthcare systems and interactions with specific health care providers within the study context. 

Response 2: Thank you for pointing this out that it was not clear how the aim of this sub-study was different from the aim of the primary study which is being published separately along with results from a randomized controlled trial with HIV self-testing kits. We have now clarified that this paper focusses on a series of questions that were specifically asking about feasibility and acceptability of sexual health counselling which were included in the focus group interview guides to specifically inform implementation and enabling environments for BSC. The Methods/Overview section has been revised as follows (page 7-8, lines 182-196): 

“The study reported here used a qualitative study design with focus groups of refugee youth conducted from August 21-27 2021 and findings from a thematic analysis are presented. It was a sub-study of the primary study called Todurujo na Kadurok (Empowering Youth), which was an exploratory sequential mixed-methods design that started with a qualitative phase to inform a randomized controlled trial with HIV self-testing kits for refugee youth[31]. The primary aim of the qualitative phase of the Todurujo na Kadurok (Empowering Youth) study was to understand HIV testing needs and experiences. Additionally, within the qualitative phase of Todurujo na Kadurok, we included a series of questions used in previous research [17,23,32] to better understand the feasibility and acceptability of sexual health counselling in clinics to inform implementation and enabling environments for BSC, results from which are presented in this paper.”

Comment 3: Further, the authors aim to explore considerations for implementing BSC, the findings presented do not adequately present contextual considerations which are transferable to aspects or components of BSC specifically. For example, could findings around relationships built between providers and youth contribute towards "reciprocal and bidirectional conversations" required in BSC? Does information sourced from family and peers contribute to "recognition of user's autonomy?". How were the findings around time to provide counselling related to feasibility of the opportunistic nature of counselling BSC offers in settings of uncertainty such as refugee camps. The authors have rightly acknowledged the limitations of their data and analysis. However, given that the title, abstract and introduction set the study's focus on BSC, there is need to show how the study findings contribute to BSC rather than sexual health counselling and services generally provided. Overall, the study does not adequately or clearly present findings within considerations for BSC. Perhaps the authors can consider how best to present findings or the discussion to draw out more clearly how the results take into consideration aspects of BSC.

Response 3: Thank you very much for this feedback. As clarified in the response above, the primary aim of this sub-study was to better understand the feasibility and acceptability of sexual health counselling in clinics, which could be used to inform implementation and enabling environments for BCS. So because we used socio-contextual theory to analyze the data we kept the results organized using relational, symbolic and material contexts that provide information about how refugee youth in Bidi Bidi (a refugee settlement context) engage with sexual health services and have conversations about sexual health, which ultimately can inform BSC implementation. Therefore, we have focussed the Discussion section – as suggested by the reviewer – to draw on how different aspects of the contexts (from the results) relate to specific aspects of BSC. Changes can be found at lines 582-584, 597-600, 605-609, 631-633, 636-648, 652-671.

Comment 4: Most of the findings also are related to doctors as the healthcare providers, have the authors considered the role of other providers in this setting (nurses and sexual health officers) towards feasibility and implementation of BSC in refugee and humanitarian settings?

Response 4: Thank you for this important question. We recognize the importance of other possible providers in the implementation of BSC in humanitarian settings so have added a section to the Results on Relational contexts about additional community resources that are used and trusted by refugee youth in Bidi Bidi as sources of sexual health information (page 15-16, lines 405-413):

“Additional access through community resources: Refugee youth also identified additional community resources where they can access health care and talk about sexual health. One refugee youth said “if I don’t go to the doctor there are also health workers who work in the community and I can go to them and explain my condition they can help me (FG2, ID5)” and another said “for cases of rape and defilement I can go the chairman of the village and share with him (FG2, ID3).” Multiple refugee youth brought up accessing sexual health care through Village Health Teams (VHTs) which are volunteers who provide basic health care services to local communities: “for me I would communicate through the VHT that I trust” (FG3, ID6).” 

We also added that young women refugees also seek care from nurses who are women if doctors who are women are not available, demonstrating the importance of women-centred care (page 20, lines 508-511): 

“They also would rather speak to a nurse who is a woman if a doctor who is a woman is not available: “Topics about menstruation I really feel bad if a male doctor starts discussing with me about menstruation. At least there should be a female doctor if not a female nurse.” (FG3, ID4)”

Comment 5: With regards stigma towards HIV, the findings seem to be presented as beliefs of healthcare staff and practitioners rather than perceptions of youth about healthcare practioners. There may be a need to clarify how these findings are presented.

Response 5: Thank you for this comment, we can see how it might have come across that the data was based on beliefs of healthcare staff but we have clarified the text to ensure it reflects the feelings of the youth: “Participants described that they often feel stigmatized by healthcare staff and practitioners when they talk about their sexual behaviour.” (page 16, lines 419-421) and “Other participants referred to feeling irresponsible or not trusted with sexual health information in the community” (page 16, lines 429-430).

It’s important to point out that refugee youth felt that healthcare staff were stigmatizing them for their sexual behaviours, but you are right, we do not have the perspectives from the healthcare providers, so we have added that as a limitation to this study: “Due to the scope of the primary study, only refugee youth were included in focus groups; however, to fully understand the extent of barriers and facilitators to sexual health counselling would have required perspectives from various healthcare providers, including understanding the extent of sexual health-related stigma that exists.” (page 27, lines 677-681)

Reviewer #2: 

Comment 1: Thank you for the opportunity to review your manuscript. This important and understudied subject.

Response 1: Thank you for your careful review and comments to strengthen our paper.

Broad comments:

Comment 2: The introduction is well written and informative.

Response 2: Thank you

Comment 3: The methods section requires a complete revision for sentence structure, flow and grammar. Some paragraphs contain sentences that are repetitive or a rewording of something that has already been mentioned.

Response 3: Thank you for this suggestion. We have gone through the methods section and revised the text for flow, grammar and deduplication. To note, you can see in the Track Changes version what was deleted. 

Comment 4: The content of the methods section requires substantial revisions. Please use some kind of qualitative reporting tool (such as the COREQ) to check your basic qualitative reporting. What is the research design? Why do you use an interview guide for a focus group discussion?

Response 4: Thank you for asking these important questions and for the suggestion to use the COREQ reporting tool, which we have added as a supplemental file. We have clarified the study design within the Overview section of the Methods as follows (page 7, lines 182-196): 

“The study reported here used a qualitative study design with focus groups of refugee youth conducted from August 21-27 2021 and findings from a thematic analysis are presented. It was a sub-study of the primary study called Todurujo na Kadurok (Empowering Youth), which was an exploratory sequential mixed-methods design that started with a qualitative phase to inform a randomized controlled trial with HIV self-testing kits for refugee youth[31]. The primary aim of the qualitative phase of the Todurujo na Kadurok (Empowering Youth) study was to understand HIV testing needs and experiences. Additionally, within the qualitative phase of Todurujo na Kadurok, we included a series of questions used in previous research [17,23,32] to better understand the feasibility and acceptability of sexual health counselling in clinics to inform implementation and enabling environments for BSC, results from which are presented in this paper.“

Also, the qualitative phase of the study used focus groups and we have corrected the text of the Methods/Data collection section to specify that we used a focus group guide to facilitate the discussions (page 9, lines 238-240):

“We used a semi-structured focus group guide to facilitate the focus group discussions. As part of the development of the focus group guide, the URDMC research assistants piloted the guide and provided feedback to ensure questions were clear and relevant to the study population.”

Comment 5: Your data analysis section does not provide enough detail. Tell the reader how and why you applied the theory. How did you check the coding? Who did you discuss and refine codes with? What did this process look like? How did you use thematic analysis? Why is this the appropriate approach to use? https://www.thematicanalysis.net/understanding-ta/ As this is a deductive approach to analysis, it’s really important that you discuss rigor and reflexivity, and that you didn’t simply see what the theory already sees. There are lots of examples of how to do this. Please look at Operationalising rigor (Lincoln and Guba, 1986, Morse, 2015).

Response: 5: Thank you for these important questions about our analyses. We have added much more detail to the Methods/Data analysis section (page 10-11, lines 276-289) to show our process of coding and underlying reason/approach to thematic analysis using both inductive and deductive analysis (based on Braun V, Clarke V. Qualitative Research in Psychology. 2006): 

“We approached thematic analysis in both an inductive way such that code and theme development was directed by the content of the data and in a deductive way by relating codes to relevant themes within each context of the socio-contextual theory[28,35]. We applied this theoretically flexible approach of thematic analysis [35,36] to explore specific aspects of material, symbolic, and relational contexts that support or prevent refugee youth from engaging with sexual health services and having conversations about sexual health. We chose to use this approach due to the flexible nature, allowing us to consider how barriers and facilitators to sexual health services and conversations are impacted by different contexts. Two University of Toronto research assistants (MGL and MC) coded the data independently and discrepant codes were reviewed together with a third party (CHL) when necessary. All codes and themes were reviewed by the senior researchers (CHL, MO), with local input from URDMC research staff (SOL, NK), and expertise from agency collaborators (KB, IT).”

Results

Comment 6: There is some overlap between sections in the results which need clarification or clearer descriptions as to why they have ended up in the section they are in. For example, the content of relational contexts is all very positive, with people talking about positive interactions with health care workers. But th

---

## [Decision Letter · Decision Letter 1]

5 Sep 2024

Qualitative insights on sexual health counselling from refugee youth in Bidi Bidi Refugee Settlement, Uganda: advancing contextual considerations for brief sexuality-related communication in a humanitarian setting

PONE-D-23-42992R1

Dear Dr. Loutet,

We’re pleased to inform you that your manuscript has been judged scientifically suitable for publication and will be formally accepted for publication once it meets all outstanding technical requirements.

Kind regards,

Sebastian Suarez Fuller, PhD

Academic Editor

PLOS ONE

Additional Editor Comments (optional):

Reviewers' comments:

Reviewer's Responses to Questions

**Comments to the Author**

1. If the authors have adequately addressed your comments raised in a previous round of review and you feel that this manuscript is now acceptable for publication, you may indicate that here to bypass the “Comments to the Author” section, enter your conflict of interest statement in the “Confidential to Editor” section, and submit your "Accept" recommendation.

Reviewer #1: All comments have been addressed

2. Is the manuscript technically sound, and do the data support the conclusions?

Reviewer #1: Yes

3. Has the statistical analysis been performed appropriately and rigorously? 

Reviewer #1: N/A

4. Have the authors made all data underlying the findings in their manuscript fully available?

Reviewer #1: No

5. Is the manuscript presented in an intelligible fashion and written in standard English?

Reviewer #1: Yes

6. Review Comments to the Author

Reviewer #1: The authors have clarified the comments provided in the earlier draft, and importantly have made this sub-study distinct from the primary study where the data was collected. It is also now clearer how the findings around feasibility and acceptability of sexual health counselling present important considerations for the implementation of brief sexuality related communication. These are reflected across methods, findinsg and discussion sections.

Data was not fully provided but the reasoning plus contact of an instutution IRB contact suffices.

7. PLOS authors have the option to publish the peer review history of their article (what does this mean?). If published, this will include your full peer review and any attached files.

Reviewer #1: No

---

## [Editor Report · Acceptance letter]

12 Nov 2024

PONE-D-23-42992R1 

PLOS ONE

Dear Dr. Loutet, 

I'm pleased to inform you that your manuscript has been deemed suitable for publication in PLOS ONE. Congratulations! Your manuscript is now being handed over to our production team.

Kind regards, 

on behalf of

Dr. Sebastian Suarez Fuller 

Academic Editor

PLOS ONE